# Unveiling the Efficiency of Vermicompost Derived from Different Biowastes on Wheat (*Triticum aestivum* L.) Plant Growth and Soil Health

**Zubair Aslam [1], Safdar Bashir [2,3,*], Waseem Hassan [4], Korkmaz Bellitürk [5], Niaz Ahmad [6], Nabeel Khan Niazi [2,7], Ahsan Khan [1], Muhammad Imran Khan [2,8], Zhongbing Chen [9,*] and Mansoor Maitah [10]**

[1] Department of Agronomy, University of Agriculture Faisalabad, Faisalabad 38040, Pakistan; zauaf@hotmail.com (Z.A.); essokhan4@gmail.com (A.K.)

[2] Institute of Soil and Environmental Sciences, University of Agriculture Faisalabad, Faisalabad 38040, Pakistan; nabeelkniazi@gmail.com (N.K.N.); khanimran1173@yahoo.com (M.I.K.)

[3] Sub-campus Depalpur, University of Agriculture Faisalabad, Okara 56130, Pakistan

[4] Department of Soil and Environmental Sciences, Muhammad Nawaz Shareef University of Agriculture, Multan 60000, Pakistan; wasagr@yahoo.com

[5] Department of Soil Science and Plant Nutrition, Faculty of Agriculture, Tekirdag Namık Kemal University, 59030 Süleymanpaşa/Tekirdağ, Turkey; kbelliturk@nku.edu.tr

[6] Department of Soil Science, Bahauddin Zakariya University, Multan 60800, Pakistan; jamniaz@yahoo.com

[7] School of Civil Engineering and Surveying, University of Southern Queensland, Toowoomba 4350, Australia

[8] Department of Isotope Biogeochemistry, Helmholtz Centre for Environmental Research—UFZ, 04318 Leipzig, Germany

[9] Czech University of Life Sciences Prague, Faculty of Environmental Sciences, Kamýcká 129, 16500 Prague, Czech Republic

[10] Department of Economics, Faculty of Economics and Management, Czech University of Life Sciences in Prague, Kamýcká 129, 16500 Prague, Czech Republic; maitah@pef.czu.cz

\* Correspondence: Safdar.bashir@uaf.edu.pk (S.B.); chenz@fzp.czu.cz (Z.C.); Tel.: +92-314-6627027 (S.B.); +420-720-153-422 (Z.C.)

**Abstract:** The present study was conducted to explore the role of different types of vermicomposts (VCs) prepared from different substrates to improve soil health (physical and chemical properties) and wheat plant growth under field conditions. Different combinations of vermicompost prepared from different substrates (cow dung, paper waste, and rice straw) and inorganic fertilizers were applied in soil using wheat as a test plant. The impact of three different VCs on physico-chemical characteristics and nutrient availability in soil was evaluated to examine their efficacy in combination with chemical fertilizers. Temporal trends in vermicomposting treatments at various stages showed significant improvement in physico-chemical attributes of the VCs substrates. All the plant physiological attributes showed significant response where N: P: K 100:50:50 kg ha$^{-1}$ + 10 t ha$^{-1}$ cow dung vermicompost was applied. In addition, post-harvest analysis of soil not only revealed that different combinations of the vermicomposting treatments improved the soil health by improving the physico-chemical attributes of the soil. Conclusively, application of cow dung vermicompost along with recommended NPK not only improved crop yield, soil health, reduced insect (aphid) infestation but also fortified grains with Zn and Fe.

**Keywords:** cow dung; vermicompost; soil health; aphid; yield; paper

## 1. Introduction

The increasing population of the world demands the adoption of an extensive and intensive cropping system with higher yields [1]. This forces intensive use of agro-chemicals leading to unsustainable practices and deteriorated environment [2]. The addition of excessive chemical fertilizers led to soil toxicities and nutrient imbalance which is a major threat for sustainable production and finally the imbalance of food chain [3]. Humans, as well as livestock, are equally affected by the residues of these agrochemicals in food products [4,5].

Globally, total organic solid wastes produced from livestock, human, and crop activities are more than 38 billion m$^3$ [6]. Rice straw produced in bulk quantity and poor feed for animals having higher silica and lingo-cellulose contents; is rather difficult to manage or dispose of [7]. Most of the farmers burn these wastes in the field which causes air pollution like smog [8]. The industries and educational institutions produce large quantities of paper waste usually used in landfills. The proper recycling of agricultural and industrial waste can play an important role in improving soil health and crop productivity. Organic matter addition in temperate region is low so the addition of organic amendments is an important component for improving soil properties and sustaining the productivity of soils [9]. The rice straw, cow dung and paper waste can be a potential substrate for vermicomposting as an effective management strategy [10].

Vermicomposting is a possible option for the management of organic solid waste and stabilizing organic material through earthworms and microorganisms [11]. Vermicompost usually contains an average of 1.5–2.2% N, 1.8–2.2% P and 1.0–1.5% K. The organic carbon is ranging from 9.15 to 17.98% and contains micronutrients [12]. The organic sources of plant nutrients (farmyard manure (FYM), crop residues, household garbage, and paper waste, etc.) using earthworms to prepare vermicompost for integrated use of plant nutrient resource and sustainable crops production is of great interest for sustainable agriculture [13].

Wheat is an essential staple food among the cereals crops in Pakistan. Despite the fact that Pakistan is producing a sufficient amount of wheat (25.6 million ton), the normal yield is lower than the yield of other countries of globe [14]. There are numerous reasons for the yield gap in wheat, but the main factor is the imbalance use of nutrients [15]. Another emerging problem is increasing aphid (*Aphidoidea*) infestation. The current reason for low wheat yields in Pakistan is the attack of Russian wheat aphids (black aphids). The infestation of the Russian wheat aphid is more severe at reproductive and grain filling stages, resulting in decline of wheat yield up to 21–92% in aphid susceptible cultivars [16]. Considering the above-mentioned facts, there is dire need for integrated use of chemical and organic fertilizers, especially the use of sustainable bio-fertilizers as an alternative approach for producing and maintaining yield at an acceptable level without compromising environment.

Realizing the importance of the vermicomposting, a first field scale attempt has been made to evaluate the efficiency of integrated use of vermicompost produced from different organic wastes and chemical fertilizer on soil properties performance and yield of wheat. The other objective was to check the potential of different vermi-composts (VCs) to reduce the total input of chemical fertilizers and aphid infestation without compromising final yield. This study also presents data of physio-chemical changes occurring in vermicompost substrate material during vermicomposting at its different developmental stages.

## 2. Materials and Methods

### 2.1. Collection of Biowastes and Earthworms

Farm waste i.e., rice straw and cow dung, was collected from the Agronomic Research Area, University of Agriculture, Faisalabad, Pakistan. Paper waste (printed and non-printed) and earthworms (*Lumbricus rubillus*) were collected from Siddiqia Photo State and Botanical Garden of University of Agriculture, Faisalabad, respectively.

### 2.2. Preparation of Vermicompost

　　　Preparation of vermicomposting was carried out at Agronomic Research Area, Department of Agronomy University of Agriculture, Faisalabad, Pakistan. Compost pits (1 m length × 1 m width × 0.5 m height) were prepared in the soil. Before the addition of waste material in the pits, pre-composting was done by following the protocols of Grag and Gupta [17]. Each waste material was mixed with soil with 5:1 ratio (organic wastes: soil) in the separate buckets ($75 \times 75 \times 60$ cm$^3$: L × W × H, respectively). Triplicates of each treatment were prepared. Sprinkler application of water was done periodically to keep organic waste material moist (50–60% moisture), to avoid odor and facilitate removal of toxic gases. The whole process took 20 days to prepare a pre-composting material. After the pre-composting, the respective materials were added into the composting pits along with 150 earthworms per each composting pit [18]. The temperature for each composting pit was maintained at $25 \pm 1\ °C$ which was optimum for earthworms [19]. Water was sprinkled on regular basis in each composting pit to keep optimum moisture level and covered each pit with wet jute bags. Mixing of waste was done periodically without disturbing the compost pit. After 180 days vermicasts were collected through sieves and fully composted vermicompost was ready to use. From each treatment, samples were taken (before vermicomposting, after 90 and 180 days of vermicomposting) for the physiochemical analysis.

### 2.3. Experimental Material

　　　The present investigation was carried out at Student Research Farm, Department of Agronomy, University of Agriculture, Faisalabad, Pakistan. Composite soil samples were collected from the top (0–30 cm) soil layer of the experimental site prior to sowing. Samples were analyzed using the protocols described by Homer and Pratt [20]. The textural class of soil was sandy clay loam. The physiochemical attributes of the soil are mentioned in Table 1. The wheat cultivar, Galaxy 2013 (Ayub Agriculture Research Institute, Faisalabad, Pakistan) was used as test cultivar which was received from Wheat Research Institute, Ayub Agricultural Research Institute, Faisalabad. Nine different treatments were developed with different combinations of NPK fertilizer and vermicompost prepared from different organic wastes (Table 2). Recommended dose of NPK for wheat crop was considered as control. Experimental treatments were arranged according to randomized complete block design with three replications.

**Table 1.** Physio-chemical parameters of soil before sowing of wheat crop.

| N (%) | AP (ppm) | AK (ppm) | Zn (ppm) | Fe (ppm) | OM (%) | pH | EC (mS cm⁻¹) | BDS (g cm⁻³) | WHD (%) |
|---|---|---|---|---|---|---|---|---|---|
| $0.052 \pm 0.01$ | $5.10 \pm 0.03$ | $240.00 \pm 0.08$ | $0.53 \pm 0.05$ | $2.13 \pm 0.02$ | $1.05 \pm 0.01$ | $7.90 \pm 0.06$ | $1.88 \pm 0.03$ | $1.49 \pm 0.02$ | $60.00 \pm 0.07$ |

N = Nitrogen; AP = Available phosphorus; AK = Available potassium; Zn = Zinc; Fe = Iron; OM = Organic matter; EC = Electrical conductivity; BDS = Bulk density of soil and WHD = Water holding capacity.

**Table 2.** Experimental treatments of the study.

| Experimental Treatment | Dose of Fertilizer (kg/ha⁻¹) | | | Dose of Vermicompost (t/ha⁻¹) | | |
|---|---|---|---|---|---|---|
| | N | P | K | Rice Straw | Paper | Cow Dung |
| $T_1$ = Control(Recommended dose of NPK) | 100 | 50 | 50 | - | - | - |
| $T_2$ | 100 | 50 | 50 | 10 | - | - |
| $T_3$ | 75 | 37.5 | 37.5 | 10 | - | - |
| $T_4$ | 50 | 25 | 25 | 10 | - | - |
| $T_5$ | 100 | 50 | 50 | - | 10 | - |
| $T_6$ | 75 | 37.5 | 37.5 | - | 10 | - |
| $T_7$ | 50 | 25 | 25 | - | 10 | - |
| $T_8$ | 100 | 50 | 50 | - | - | 10 |
| $T_9$ | 75 | 37.5 | 37.5 | - | - | 10 |
| $T_{10}$ | 50 | 25 | 25 | - | - | 10 |

$T_1$ = Control (Recommended NPK 100:50:50 kg ha⁻¹); $T_2$ = N: P: K 100:50:50 kg ha⁻¹ + 10 t ha⁻¹ rice straw vermicompost; $T_3$ = N: P: K 75:37.5:37.5 kg ha⁻¹ + 10 t ha⁻¹ rice straw vermicompost; $T_4$ = N: P: K 50:25:25 kg ha⁻¹ + 10 t ha⁻¹ rice straw vermicompost; $T_5$ = N: P: K 100:50:50 kg ha⁻¹ + 10 t ha⁻¹ paper waste vermicompost; $T_6$ = N: P: K 75:37.5:37.5 kg ha⁻¹ + 10 t ha⁻¹ paper waste vermicompost; $T_7$ = N: P: K 50:25:25 kg ha⁻¹ + 10 t ha⁻¹ paper waste vermicompost; $T_8$ = N: P: K 100:50:50 kg ha⁻¹ + 10 t ha⁻¹ cow dung vermicompost; $T_9$ = N: P: K 75:37.5:37.5 kg ha⁻¹ + 10 t ha⁻¹ cow dung vermicompost and $T_{10}$ = N: P: K 50:25:25 kg ha⁻¹ + 10 t ha⁻¹ cow dung vermicompost.

## 2.4. Crop Husbandry

Wheat crop was sown on 20 November 2016. The experimental plot size was 386 m$^2$. The land was prepared by performing two cross-ploughings each followed by planking with the help of a tractor drawn tine-cultivator to achieve the normal seedbed. A hand drill was used for the sowing of seeds maintaining 23 cm row to row distance using the seed rate at 125 kg ha$^{-1}$ (recommended for normal sowing time in Punjab, Pakistan). Required amount of fertilizer dose NPK ha$^{-1}$ in the form of urea, diammonium phosphate and murate of potash (MOP), was applied. Full dose of P and K and 1/3rd of urea was applied at sowing while the remaining dose of urea applied at two critical stages as tillering and spikelets initiation. Different doses of vermicompost material were added at the time of sowing. In total, four irrigations were applied to the crop during the growth period in addition to soaking irrigation of four-acre inches. Crop was harvested on 11 April 2017, at harvest maturity.

## 2.5. Data Collection

For measuring the pH and EC of organic waste the protocols of Ryan et al. [21] were followed. For the measurement Zn and Fe content in soil, DTPA test was used as described by the Lindsay and Norvell [22]. Total nitrogen, available phosphorus, available potassium and organic matter in the organic samples were determined by following the protocols of Bremner and Mulavaney [23], Olsen and Sommers [24], Helmke and Sparks [25]and Walkely and Black [26], respectively. Data on aphid population (aphid per tiller) were recorded from twenty tillers at milking stage in each plot manually and then averaged. For chlorophyll content, leaves sample of one gram were taken from each treatment at the tillering stage and measured following the protocol of Nagata and Yamashita [27]. Plant height was measured at maturity stage with the help of meter rod starting from the base of plant up to the tip of flag leaf. Unit area was selected randomly from each plot for the counting of the number of total and productive tillers. Ten spikes were selected at random from each plot, and number of spikelets and grains in each spike were counted. The plants at maturity were harvested for biological and grain yield. The spikes were threshed manually, and the grains and plant biomass were weighed on a weighing balance. For 1000-grain weight, a sub sample of 100 grains taken from each treatment and weighed on weighing balance. The harvest index (H.I) was computed using the formula, HI = (Grain yield / Biological yield) × 100. Grain protein contents were determined by Kjeldahl digestion method while the Fe and Zn contents in the grain samples were determined by following the protocols of Jones and Case [28]. Post-harvest analysis of soil from plots of each treatment was also carried out using above mentioned protocols.

## 2.6. Statistical Analysis

The data from this experiment was analyzed using the software (STATISTIX 8.1) (Analytical software, Tallahassee, FL, USA) program. When significant differences were detected by the analysis of variance (ANOVA), Fisher's least significant difference was used to compare treatment means at $p \leq 0.05$ level of significance.

## 3. Results

Physicochemical attributes of different organic wastes at various stages of the vermicomposting significantly improved (Table 3). Maximum nitrogen contents (0.14%), available phosphorous (15.50 ppm), available potassium (1042.00 ppm), Zn contents (0.62 ppm) and EC (7.26 m S cm$^{-1}$) were observed in the cow dung followed by the rice straw and paper waste at pre-composting. Maximum concentration of Fe (4.10 ppm) and OM (3.0%) were noticed in the paper waste at first stage of vermicomposting. However, the pH showed non-significant results for all the organic wastes. Regarding second stage of the vermicomposting after 90 days that was considered as the immature stage, maximum available phosphorous (17.70 ppm), available potassium (840.00 ppm), OM (2.58%) and EC (6.89 m S cm$^{-1}$) was observed in the paper waste followed by the cow dung and rice straw while the maximum limits for the Zn (0.84 ppm) was observed in the cow dung and Fe contents (4.08

ppm) was observed in rice straw. However, the N contents and pH showed non-significant results for all the organic wastes during second stage of the vermicomposting (Table 3). After the 180 days of vermicomposting, all the physio-chemical attributes showed significant behavior for the different organic wastes except for pH. All the organic wastes showed different behavior for all the physio-chemical attributes.

**Table 3.** Physio-chemical parameters of different organic wastes at different stages of vermicomposting.

| Treatment | N (%) | AP (ppm) | AK (ppm) | Zn (ppm) | Fe (ppm) | OM (%) | pH | EC (m S cm⁻¹) |
|---|---|---|---|---|---|---|---|---|
| | | | Before Vermicomposting | | | | | |
| Paper waste | 0.11 a | 9.10 b | 280.00 c | 0.41 b | 4.10 a | 3.00 a | 8.20 | 4.87 c |
| Cow dung | 0.02 b | 15.50 a | 1042.00 a | 0.62 a | 3.84 b | 2.62 b | 8.30 | 7.26 a |
| Rice straw | 0.14 a | 14.40 a | 800.00 b | 0.58 a | 3.28 c | 2.72 b | 8.10 | 6.05 b |
| LSD ($p \leq 0.05$) | 0.05 | 1.02 | 203.23 | 0.07 | 0.49 | 0.23 | NS | 1.03 |
| | | | After 90-days of Vermicomposting (Immature) | | | | | |
| Paper waste | 0.18 | 17.70 a | 840.00 a | 0.72 b | 2.90 c | 2.58 a | 8.00 | 6.89 a |
| Cow dung | 0.15 | 14.90 b | 760.00 b | 0.84 a | 3.54 b | 2.37 b | 8.00 | 6.45 b |
| Rice straw | 0.12 | 10.80 c | 360.00 c | 0.70 b | 4.08 a | 2.30 b | 7.90 | 3.42 c |
| LSD ($p \leq 0.05$) | NS | 2.56 | 243.23 | 0.05 | 0.34 | 0.14 | NS | 0.21 |
| | | | After 180-days of Vermicomposting (Mature) | | | | | |
| Paper waste | 0.24 a | 23.21 a | 1425.00 a | 1.02 b | 1.83 c | 2.12 a | 8.01 | 6.12 a |
| Cow dung | 0.30 a | 11.56 b | 346.00 b | 1.06 a | 3.09 b | 2.10 a | 8.00 | 5.95 a |
| Rice straw | 0.07 b | 6.34 c | 127.00 c | 0.97 c | 4.21 a | 2.01 b | 7.98 | 2.98 b |
| LSD ($p \leq 0.05$) | 0.12 | 3.45 | 234.39 | 0.03 | 0.56 | 0.04 | NS | 1.23 |

N = Nitrogen; AP = Available phosphorus; AK = Available potassium; Zn = Zinc; Fe = Iron; OM = Organic matter and EC = Electrical conductivity. Any two means within a column followed by same letters are not significant at p ≤ 0.05. n = 3. NS = non-significant.

All the vermicompost treatments significantly improved the yield attributes of the wheat cultivar, Galaxy 2013. The maximum plant height (97.2 cm), total tillers (439.33 m$^{-2}$), productive tillers (428.33 m$^{-2}$), chlorophyll contents (5.10 mg L$^{-1}$), spike length (11.0 cm), spiklets per spike (20.00), grains per spike (52.00), 1000-grains weight (38.76 g), grain yield (5.37 t ha$^{-1}$), biological yield (12.06 t ha$^{-1}$) and harvest index (41.32%) were observed where N: P: K 100:50:50 kg ha$^{-1}$ + 10 t ha$^{-1}$ cow dung vermicompost was applied followed by in that treatment where N: P: K 100:50:50 kg ha$^{-1}$ + 10 t ha$^{-1}$ paper waste vermicompost was applied. However, minimum plant height (75.3 cm), total tillers (397.00 m$^{-2}$), productive tillers (381.00 m$^{-2}$), chlorophyll contents (3.92 mg L$^{-1}$), spike length (8.1 cm), spiklets per spike (16.00), grains per spike (39.00), 1000-grains weight (26.26 g), grain yield (3.22 t ha$^{-1}$), biological yield (8.15 t ha$^{-1}$) and harvest index (26.67%) were observed where N: P: K 50:25:25 kg ha$^{-1}$ + 10 t ha$^{-1}$ paper waste vermicompost was applied (Tables 4 and 5).

**Table 4.** Influence of different vermicompost (mature) treatments on the yield related attributes of wheat.

| Treatment | Plant Height (cm) | Total Tillers (m⁻²) | Productive Tillers (m⁻²) | Spike Length (cm) | Spikelets /Spike | Grains /Spike | 1000-Grains Weight (g) | Biological Yield (t ha⁻¹) | Grain Yield (t ha⁻¹) | Harvest Index (%) |
|---|---|---|---|---|---|---|---|---|---|---|
| T$_1$ | 88.2 DE | 423.67 B–D | 406.67 C–D | 10.0 CD | 17.66 C–E | 47.33 BC | 32.23 C–E | 10.53 BC | 4.81 B–D | 36.76 C |
| T$_2$ | 95.6 AB | 436.67 AB | 425.67 A | 10.8 AB | 19.33 AB | 50.66 AB | 37.53 AB | 11.98 A | 5.30 AB | 39.25 B |
| T$_3$ | 85.3 EF | 421.33 CD | 406.67 B–D | 9.8 CD | 17.00 D–F | 46.33 CD | 30.02 D–F | 9.76 CD | 4.74 CD | 34.83 D |
| T$_4$ | 78.5 HI | 403.67 EF | 387.67 FG | 8.3 F | 16.66 EF | 40.00 FG | 27.86 F | 8.26 F | 3.78 E | 28.71 G |
| T$_5$ | 93.4 BC | 431.82 A–C | 420.67 AB | 10.4 A–C | 18.66 BC | 48.33 BC | 35.16 A–C | 11.80 A | 5.11 A-C | 38.73 B |
| T$_6$ | 82.3 FG | 418.32 CD | 402.33 D-E | 9.4 DE | 18.00 CD | 43.66 DE | 29.56 EF | 9.43 DE | 4.62 CD | 32.88 E |
| T$_7$ | 75.3 I | 397.00 F | 381.00 G | 8.1 F | 16.00 F | 39.00 G | 26.26 F | 8.15 F | 3.22 F | 26.67 H |
| T$_8$ | 97.2 A | 439.33 A | 428.33 A | 11.0 A | 20.00 A | 52.00 A | 38.76 A | 12.06 A | 5.37 A | 41.32 A |
| T$_9$ | 90.3 CD | 426.47 A–D | 410.33 B–C | 10.1BC | 18.00 CD | 48.33 BC | 33.93 B–D | 10.93 B | 5.04 A–C | 37.19 C |
| T$_{10}$ | 80.4 GH | 412.33 DE | 396.33 EF | 9.0 E | 17.00 D–F | 42.66 EF | 28.89 EF | 8.63 EF | 4.30 D | 30.89 F |
| LSD value | 3.47 | 14.36 | 15.03 | 0.72 | 1.12 | 3.47 | 4.06 | 0.83 | 0.50 | 1.34 |

T$_1$ = Control (Recommended NPK 100:50:50 kg ha⁻¹); T$_2$ = N: P: K 100:50:50 kg ha⁻¹ + 10 t ha⁻¹ rice straw vermicompost; T$_3$ = N: P: K 75:37.5:37.5 kg ha⁻¹ + 10 t ha⁻¹ rice straw vermicompost; T$_4$ = N: P: K 50:25:25 kg ha⁻¹ + 10 t ha⁻¹ rice straw vermicompost; T$_5$ = N: P: K 100:50:50 kg ha⁻¹ + 10 t ha⁻¹ paper waste vermicompost; T$_6$ = N: P: K 75:37.5:37.5 kg ha⁻¹ + 10 t ha⁻¹ paper waste vermicompost; T$_7$ = N: P: K 50:25:25 kg ha⁻¹ + 10 t ha⁻¹ paper waste vermicompost; T$_8$ = N: P: K 100:50:50 kg ha⁻¹ + 10 t ha⁻¹ cow dung vermicompost; T$_9$ = N: P: K 75:37.5:37.5 kg ha⁻¹ + 10 t ha⁻¹ cow dung vermicompost and T$_{10}$ = N: P: K 50:25:25 kg ha⁻¹ + 10 t ha⁻¹ cow dung vermicompost. Any two means within a column followed by same letters are not significant at $p \le 0.05$. n = 3.

Data regarding the quality attributes of wheat documented in Table 5 showed that all the vermicompost treatments significantly improved the quality attributes of the wheat cultivar, Galaxy 2013. The maximum grain Zn contents (24.37 ppm), grain Fe contents (34.63 ppm) and grain protein contents (15.37%) were observed where N: P: K 100:50:50 kg ha$^{-1}$ + 10 t ha$^{-1}$ cow dung vermicompost was applied followed by treatment N: P: K 100:50:50 kg ha$^{-1}$ + 10 t ha$^{-1}$ paper waste vermicompost. However minimum grain Zn contents (17.47 ppm), grain Fe contents (24.18 ppm) and grain protein contents (13.50%) was observed where N: P: K 50:25:25 kg ha$^{-1}$ + 10 t ha$^{-1}$ paper waste vermicompost was applied (Table 5).

**Table 5.** Influence of different vermicompost treatments on the biochemical attributes of wheat.

| Treatment | Chlorophyll Contents (mg L$^{-1}$) | Grain Zinc Contents (ppm) | Grain Iron Contents (ppm) | Grain Protein Contents (%) |
|---|---|---|---|---|
| T$_1$ | 4.65 DE | 21.91 C | 30.09 D | 15.03 D |
| T$_2$ | 4.94 B | 23.30 B | 32.63 B | 15.69 B |
| T$_3$ | 4.65 DE | 20.21 D | 27.18 F | 14.71 E |
| T$_4$ | 4.11 H | 18.61 F | 25.19 G | 12.93 I |
| T$_5$ | 4.85 BC | 22.54 C | 31.37 C | 15.55 B |
| T$_6$ | 4.46 FG | 20.45 D | 28.48 E | 14.42 F |
| T$_7$ | 3.92 I | 17.47 G | 24.18 H | 13.50 H |
| T$_8$ | 5.10 A | 24.37 A | 34.63 A | 15.97 A |
| T$_9$ | 4.73 CD | 22.08 C | 31.11 C | 15.29 C |
| T$_{10}$ | 4.37 G | 19.46 E | 25.78 G | 14.09 G |
| LSD value | 0.13 | 0.65 | 1.01 | 0.19 |

T$_1$ = Control (Recommended NPK 100:50:50 kg ha$^{-1}$); T$_2$ = N: P: K 100:50:50 kg ha$^{-1}$ + 10 t ha$^{-1}$ rice straw vermicompost; T$_3$ = N: P: K 75:37.5:37.5 kg ha$^{-1}$ + 10 t ha$^{-1}$ rice straw vermicompost; T$_4$ = N: P: K 50:25:25 kg ha$^{-1}$ + 10 t ha$^{-1}$ rice straw vermicompost; T$_5$ = N: P: K 100:50:50 kg ha$^{-1}$ + 10 t ha$^{-1}$ paper waste vermicompost; T$_6$ = N: P: K 75:37.5:37.5 kg ha$^{-1}$ + 10 t ha$^{-1}$ paper waste vermicompost; T$_7$ = N: P: K 50:25:25 kg ha$^{-1}$ + 10 t ha$^{-1}$ paper waste vermicompost; T$_8$ = N: P: K 100:50:50 kg ha$^{-1}$ + 10 t ha$^{-1}$ cow dung vermicompost; T$_9$ = N: P: K 75:37.5:37.5 kg ha$^{-1}$ + 10 t ha$^{-1}$ cow dung vermicompost and T$_{10}$ = N: P: K 50:25:25 kg ha$^{-1}$ + 10 t ha$^{-1}$ cow dung vermicompost. Any two means within a column followed by same letters are not significant at $p \leq 0.05$. n = 3.

During the study, aphid population was also studied for each treatment. Figure 1 showed that maximum aphid population was observed in the treatment where N: P: K 50:25:25 kg ha$^{-1}$ + 10 t ha$^{-1}$ paper waste vermicompost was applied while minimum aphid population was observed where N: P: K 100:50:50 kg ha$^{-1}$ + 10 t ha$^{-1}$ cow dung vermicompost was applied.

Economic analysis provides a better way to predict the performance of better treatment. Regarding the economic analysis, maximum benefit cost ratio (BCR) (1.20) was recorded in the treatment where N: P: K 100:50:50 kg ha$^{-1}$ + 10 t ha$^{-1}$ cow dung vermicompost was applied (Table 6). However, minimum BCR was observed in T$_7$ where N: P: K 50:25:25 kg ha$^{-1}$ + 10 t ha$^{-1}$ paper waste vermicompost was applied (Table 6).

**Table 6.** Economic analysis of wheat grown in various chemical fertilizer and vermicompost treatments.

| Treatment | Yield (kg ha⁻¹) | Adjusted Yield (kg ha⁻¹) | Gross Income (US$ ha⁻¹) | Total Cost (US$ ha⁻¹) | Net Benefits (US$ ha⁻¹) | Benefit Cost Ratio |
|---|---|---|---|---|---|---|
| $T_1$ | 4810 | 4329 | 1365.94 | 672.65 | 693.29 | 1.03 |
| $T_2$ | 5300 | 4770 | 1505.09 | 695.30 | 809.78 | 1.16 |
| $T_3$ | 4740 | 4266 | 1346.08 | 658.35 | 687.70 | 1.04 |
| $T_4$ | 3780 | 3402 | 1073.44 | 621.44 | 451.99 | 0.72 |
| $T_5$ | 5110 | 4599 | 1451.13 | 816.64 | 634.49 | 0.77 |
| $T_6$ | 4620 | 4158 | 1311.99 | 779.69 | 532.29 | 0.68 |
| $T_7$ | 3220 | 2898 | 914.41 | 742.78 | 171.62 | 0.23 |
| $T_8$ | 5370 | 4833 | 1524.97 | 690.45 | 834.51 | 1.20 |
| $T_9$ | 5040 | 4536 | 1431.26 | 653.50 | 777.74 | 1.19 |
| $T_{10}$ | 4300 | 3870 | 1221.11 | 616.59 | 604.51 | 0.98 |
| Remarks | US$ 0.29/kg | 10% less than actual | (US$ ha⁻¹) | (US$ ha⁻¹) | (US$ ha⁻¹) | |

$T_1$ = Control (Recommended NPK 100:50:50 kg ha⁻¹); $T_2$ = N: P: K 100:50:50 kg ha⁻¹ + 10 t ha⁻¹ rice straw vermicompost; $T_3$ = N: P: K 75:37.5:37.5 kg ha⁻¹ + 10 t ha⁻¹ rice straw vermicompost; $T_4$ = N: P: K 50:25:25 kg ha⁻¹ + 10 t ha⁻¹ rice straw vermicompost; $T_5$ = N: P: K 100:50:50 kg ha⁻¹ + 10 t ha⁻¹ paper waste vermicompost; $T_6$ = N: P: K 75:37.5:37.5 kg ha⁻¹ + 10 t ha⁻¹ paper waste vermicompost; $T_7$ = N: P: K 50:25:25 kg ha⁻¹ + 10 t ha⁻¹ paper waste vermicompost; $T_8$ = N: P: K 100:50:50 kg ha⁻¹ + 10 t ha⁻¹ cow dung vermicompost; $T_9$ = N: P: K 75:37.5:37.5 kg ha⁻¹ + 10 t ha⁻¹ cow dung vermicompost and $T_{10}$ = N: P: K 50:25:25 kg ha⁻¹ + 10 t ha⁻¹ cow dung vermicompost. 1 US$ = 103 PKR (In April 2017).

Post-harvest analysis of each treatment showed that application of N: P: K 100:50:50 kg ha$^{-1}$ + 10 t ha$^{-1}$ cow dung vermicompost improved the physio-chemical attributes of the soil. Maximum improvement in the N contents ($0.25 \pm 0.01\%$), available phosphorus ($6.01 \pm 0.04$ ppm), available potassium ($321.11 \pm 0.04$ ppm), Zn contents ($0.73 \pm 0.01$ ppm), Fe contents ($251.82 \pm 0.04$ ppm), OM ($1.08 \pm 0.02\%$) and water holding capacity ($67.03 \pm 0.02\%$) was noticed in $T_8$ where N: P: K 100:50:50 kg ha$^{-1}$ + 10 t ha$^{-1}$ cow dung vermicompost was applied. Minor change in the pH, EC and bulk density of soil was noticed in the same treatment.

## 4. Discussion

The present study revealed that management of different wastes by vermi-technology can play a vital role in sustainable agriculture. The present investigation shows the effective transformation of farm wastes, household/office wastes and livestock waste into a valuable product which can be used for sustainable agriculture. These wastes cause severe disposal and environmental problem [29]. Not only have we proved this in our study, but other investigations showed that vermicomposting can play a vital role in sustainable farming by converting natural and anthrophonic wastes in to quality organic manure. Not only this study, but also various other investigations prove that agriculture wastes, household and livestock wastes can be efficiently utilized using this technology not only as an alternative nutrient source but also to improve the physical, chemical and bio-logical properties of soil [30–32].

This study also shows that physio-chemical parameters (pH, EC and OM contents) along with the concentration of macro (AP, N, AK) and micro (Fe and Zn) contents of nutrients of different organic wastes (cow dung, paper waste, and rice straw) at various stages of vermicomposting were significantly influenced. With the passage of time, all the physiochemical attributes along with the concentration of micro and macro nutrients were changed (Table 3). The variable rate of bioconversion of all the organic wastes (paper waste, cow dung and rice straw) significantly affected physico-chemical characteristics vermicompost. The findings of the present study are similar to that of Suthar, S. [33] who reported the different conversion rates of some organic wastes when subjected to vermicompost. It was found that earthworms first choose easily decomposable substrates with more available nutrients. Different studies reported the changes in pH and EC during vermicomposting process for different organic wastes. Our results are similar to the findings of Fares et al. [34], which showed an increase in pH during the conversion of crop residues and animal manures to vermicompost. This increase in pH of vermicompost might be attributed to the release of $NH_4^+$ ions that ultimately reduced $H^+$. In the case of EC, previous studies found an increase in EC during vermicomposting and composting processes [10,35,36] which is similar to our findings. However, in case of wheat straw the EC decreased during vermicomposting which is similar to the findings of Panjgotra et al. and this might be due to loss of organic matter during vermicomposting. The rise in EC values might be attributed due to the release of minerals such as exchangeable K, Ca, Mg, and P in the available forms from the decomposing organic substrate in the vermicompost.

The findings of this study also showed that organic matter percentage is highly influenced during the vermicomposting process due to chemical characteristics of the initial substrates (paper waste, cow dung and rice straw) used for decomposition (Table 3). Earlier findings of the Raghavendra and Bano also reported that organic matter percentage is highly related to chemical composition of the substrate [37]. Thus, the composition of substrate is a major factor which determines the efficiency of vermicomposting. Vermicomposting is a process of bio-oxidation and stabilization of organic material, which is different from composting as it involves the joint action of earthworms and microorganisms. Various stages of vermicomposting significantly reduced the percentage of organic matter percentage for all the organic wastes (Table 3). Mineralization mediated by microorganisms and earthworms results in the decrease in organic matter during vermicomposting process. Catabolic action of earthworms modifies the substrate condition, which consequently increases the surface area of substrate material for microbial action [38], which ultimately promotes carbon loss through microbial respiration. Different wastes have different rates of loss of OM depending upon the composition. Organic-C losses has been found to reach 52% in

poultry manure, 67% in cattle manure and 72% in pig manure during composting. [39]. Biotransformation through earthworms resulted in the synthesis of quality vermicomposts with significant levels of plant available nutrients. When the organic wastes get transformed through vermi-technology, substantial quantitative improvements in micro and macro nutrients were detected, making vermi-composts as an important organic sources of nutrients for crop production. We also found nitrogen, available phosphorus, available potassium, Zn and Fe contents ranging from (0.02–0.30%), (9.10–23.21 ppm), (127.00–1425.00 ppm), (0.41–1.06 ppm) and (1.83–4.21 ppm), respectively, for all the organic wastes (paper wastes, cow dung and rice straw) at various stages of vermicomposting (Table 3). Our findings are correlated with the findings of Bansal and Kapoor who observed an increase in the concentration of macro and micronutrients on vermicomposting of mustard residue and sugarcane thrash [40]. The present results are also similar to that of Chauhan and Joshi, who reported considerable rise in nutrient behavior in vermi-composts of some weeds such as congress grass (*Parthenium hysterophorus*), water hyacinth (*Eichhornia crassipes*) and bhang (*Cannabis sativa*) [41]. Nutrient behavior in vermicompost is mainly affected by the nutrient content of the organic waste used as earthworm feed [42]. Earthworms has a crucial role in increasing and improving the nitrogen contents of the waste by adding nitrogen rich mucus, decaying tissues of dead worms and by enhancing microbial mediated nitrogen mineralization [33]. Phosphorous mineralization during the vermicomposting process results in enhanced available phosphorous level in vermicompost [43] which might be due to the action of earthworms' phosphatases and P-solubilizing microorganisms in the earthworm gut [44]. The rise in available potassium (AK) concentrations in vermicompost as compared to that of standard composts and biomass might be due to physical enzymatic action and grinding during the passage of substrate through the gut [45]. In contrast to these findings, some studies also reported the reduced levels of AK in vermicompost as compared to substrate material. This perhaps reflects leaching of this soluble element by the excessive water which might be drained from the composting material [46].

Efficacy of vermicomposts prepared from different organic wastes combined with inorganic fertilizer in terms of promoting plant growth was investigated in this study. All the vermicompost treatments significantly improved the yield attributes of the wheat cultivar, Galaxy 2013. The maximum of all the yield attributes were observed where N: P: K 100:50:50 kg ha$^{-1}$ + 10 t ha$^{-1}$ cow dung vermicompost was applied (Table 4). It was noticed during this investigation that application of vermicompost, with or without fertilizer, significantly improved the yield attributes of different cereals, pulses, oil seed crops, spices, vegetables, and varieties of fruit trees [47]. Strategic planning in terms of the integrated application of manures with inorganic fertilizers can sustain the soils and benefit the farmers and our results are in accordance with this statement [48]. Nutrient uptake is better in crops when inorganic fertilizers are used with organic manure products like vermicompost compared to the application of inorganic fertilizers alone [49]. These results are in line with the findings of many researchers who reported that combined application of manures and fertilizers increased the plant height and tillers hill$^{-1}$ [50], spike length [51] and filled grains spike$^{-1}$ [52], grain weight and finally the yield of the wheat crop [53]. Similarly, Dynes examined the impact of vermicompost compared with the industrial compost and NPK fertilizer on growth and yield of cucumber and reported that vermicompost increased plant efficiency [54]. They also declared that the application of vermicompost mixed with topsoil has a positive effect on growth of cucumber plants and we found similar results in case of wheat.

Different vermicompost treatments significantly improved the biochemical and quality attributes of the wheat cultivar. Maximum chlorophyll contents, grain Zn, Fe and protein contents were observed where N: P: K 100:50:50 kg ha$^{-1}$ + 10 t ha$^{-1}$ cow dung vermicompost was applied (Table 5). Similarly, minimum aphid population was also noticed in the same treatment (Figure 1). Leventoglu and Erdal, stated that higher levels of organic matters can bind soil nutrients as unavailable forms, thus plants cannot grow better in the terms of yield and quality traits [55]. Due to large particulate surface areas, vermi-composts provide many micro sites for microbial activity and for the strong retention of nutrients [56]. These results are also supported by other researchers who recorded that there had been some growth improving products such as hormone like substances,

cytokinins, auxins and humates produced with some microorganism and earthworms [57]. Vermicompost contains most nutrients in plant-available forms that ultimately enhance the biochemical, yield and quality attributes of the crops [10,45]. These properties of vermicomposts might be the reason for the improving of the quality of the final product as we have observed in our study. Also, increasing effect of vermicompost on soil nutrient availability might lead to an increase in plant mineral nutrition. Regarding the benefit cost ratios (BCR), maximum outcomes were observed with the vermicompost obtained from the cow dung because of higher nutritional outcomes and the easily availability at the lower cost and with higher final yield and quality. All the vermicompost treatments significantly improved the physio-chemical parameters of soil after harvesting of wheat crop (Table 7). These results are supported by the findings of Sable et al. who stated that maximum yields and availability of nutrients in soil after harvest was achieved in a treatment where 50% of N was supplied by vermicompost and 50% by neem cake [58]. Similarly, the overall performance of crop and postharvest physio-chemical parameters of soil were found to be better when the required inorganic fertilizer was reduced to 50% of the recommended level and applied together with 5–10 ton ha$^{-1}$ of vermicompost for any crop [59,60]. These results are supported by the findings of Sreenivas et al. who stated that application of vermi composts into soils significantly improved the post-harvest physio- chemical attributes of the soil [61]. So, we have to notice that application of N: P: K 100:50:50 kg ha$^{-1}$ + 10 t ha$^{-1}$ cow dung vermicompost significantly improved the yield and quality attributes of wheat with improving its grain biofortification and ultimately the health of soil. The results of this study can be recommended for different crops as vermicomposts from different feeding stocks have different physio-chemical compositions and they can be recommended to reduce the recommended dose of chemical fertilizers for economic and environmental point of view. There is still need to explore the efficacy of vermicompost without inorganic fertilizer and mechanistic investigations of vermicomposts as biocontrols for aphids in wheat and other crops.

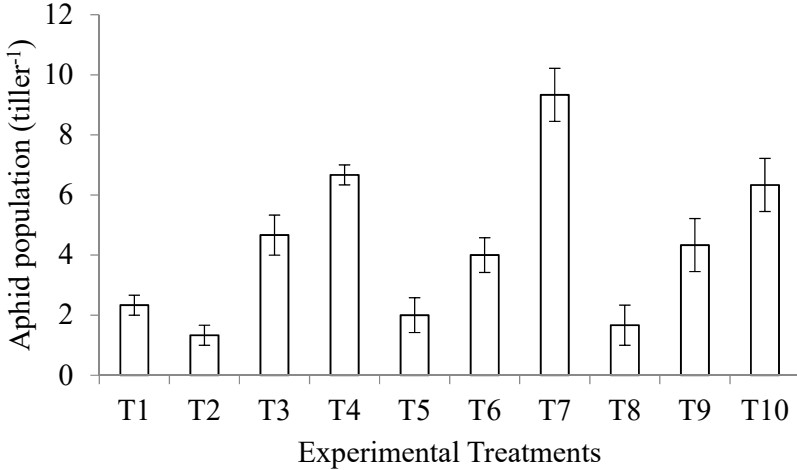

**Figure 1.** Aphid population as affected by the use various fertilizer and vermi-compost treatments.

**Table 7.** Physio-chemical parameters of soil after harvesting of wheat crop.

| Treatment | N (%) | AP (ppm) | AK (ppm) | Zn (ppm) | Fe (ppm) | OM (%) | pH | EC (mS cm$^{-1}$) | BDS (g cm$^{-3}$) | WHD (%) |
|---|---|---|---|---|---|---|---|---|---|---|
| $T_1$ | 0.17 ± 0.01 | 5.83 ± 0.03 | 294.31 ± 0.10 | 0.66 ± 0.01 | 244.67 ± 0.10 | 1.07 ± 0.01 | 7.48 ± 0.01 | 1.59 ± 0.01 | 1.66 ± 0.02 | 66.51 ± 0.10 |
| $T_2$ | 0.21 ± 0.03 | 5.96 ± 0.01 | 312.78 ± 0.11 | 0.70 ± 0.02 | 250.01 ± 0.11 | 1.08 ± 0.02 | 7.42 ± 0.02 | 1.53 ± 0.01 | 1.68 ± 0.03 | 66.91 ± 0.09 |
| $T_3$ | 0.14 ± 0.02 | 5.80 ± 0.01 | 291.87 ± 0.05 | 0.62 ± 0.01 | 242.89 ± 0.12 | 1.06 ± 0.01 | 7.50 ± 0.01 | 1.57 ± 0.02 | 1.64 ± 0.03 | 66.34 ± 0.05 |
| $T_4$ | 0.08 ± 0.01 | 5.71 ± 0.03 | 275.34 ± 0.09 | 0.61 ± 0.01 | 241.10 ± 0.07 | 1.03 ± 0.02 | 7.46 ± 0.01 | 1.64 ± 0.01 | 1.60 ± 0.02 | 65.99 ± 0.03 |
| $T_5$ | 0.19 ± 0.01 | 5.91 ± 0.01 | 309.45 ± 0.10 | 0.68 ± 0.02 | 247.97 ± 0.05 | 1.07 ± 0.04 | 7.43 ± 0.01 | 1.54 ± 0.01 | 1.62 ± 0.04 | 66.78 ± 0.04 |
| $T_6$ | 0.12 ± 0.01 | 5.77 ± 0.02 | 286.56 ± 0.04 | 0.63 ± 0.03 | 243.12 ± 0.03 | 1.05 ± 0.02 | 7.46 ± 0.02 | 1.60 ± 0.01 | 1.64 ± 0.05 | 66.23 ± 0.05 |
| $T_7$ | 0.08 ± 0.01 | 5.70 ± 0.02 | 261.89 ± 0.05 | 0.60 ± 0.02 | 240.53 ± 0.02 | 1.02 ± 0.01 | 7.45 ± 0.04 | 1.67 ± 0.01 | 1.62 ± 0.03 | 65.78 ± 0.03 |
| $T_8$ | 0.25 ± 0.01 | 6.01 ± 0.04 | 321.11 ± 0.04 | 0.73 ± 0.01 | 251.82 ± 0.04 | 1.08 ± 0.02 | 7.40 ± 0.01 | 1.50 ± 0.03 | 1.70 ± 0.02 | 67.03 ± 0.02 |
| $T_9$ | 0.16 ± 0.02 | 5.87 ± 0.01 | 304.34 ± 0.05 | 0.66 ± 0.01 | 247.78 ± 0.05 | 1.05 ± 0.01 | 7.48 ± 0.01 | 1.55 ± 0.02 | 1.67 ± 0.01 | 66.63 ± 0.04 |
| $T_{10}$ | 0.09 ± 0.03 | 5.73 ± 0.01 | 281.92 ± 0.07 | 0.64 ± 0.02 | 242.02 ± 0.05 | 1.04 ± 0.01 | 7.43 ± 0.01 | 1.63 ± 0.01 | 1.63 ± 0.02 | 66.09 ± 0.04 |

N = Nitrogen; AP = Available phosphorus; AK = Available potassium; Zn = Zinc; Fe = Iron; OM = Organic matter; EC = Electrical conductivity; BDS = Bulk density of soil and WHD = Water holding capacity. $T_1$ = Control (Recommended NPK 100:50:50 kg ha$^{-1}$); $T_2$ = N: P: K 100:50:50 kg ha$^{-1}$ + 10 t ha$^{-1}$ rice straw vermicompost; $T_3$ = N: P: K 75:37.5:37.5 kg ha$^{-1}$ + 10 t ha$^{-1}$ rice straw vermicompost; $T_4$ = N: P: K 50:25:25 kg ha$^{-1}$ + 10 t ha$^{-1}$ rice straw vermicompost; $T_5$ = N: P: K 100:50:50 kg ha$^{-1}$ + 10 t ha$^{-1}$ paper waste vermicompost; $T_6$ = N: P: K 75:37.5:37.5 kg ha$^{-1}$ + 10 t ha$^{-1}$ paper waste vermicompost; $T_7$ = N: P: K 50:25:25 kg ha$^{-1}$ + 10 t ha$^{-1}$ paper waste vermicompost; $T_8$ = N: P: K 100:50:50 kg ha$^{-1}$ + 10 t ha$^{-1}$ cow dung vermicompost; $T_9$ = N: P: K 75:37.5:37.5 kg ha$^{-1}$ + 10 t ha$^{-1}$ cow dung vermicompost and $T_{10}$ = N: P: K 50:25:25 kg ha$^{-1}$ + 10 t ha$^{-1}$ cow dung vermicompost.

## 5. Conclusions

Due to an increase in available macro- and micro-nutrients, vermicompost are rich source of nutrients and they can be easily used in combination with chemical fertilizers to reduce the recommended dose, as well as being the best nutritional source of biofortification. Vermicompost can be a biocontrol agent for aphid attack indirectly due to synthesis of phytohormones, the increase in nitrogen and phosphorous uptake and the increase in iron and mineral solubility through chelation growth. The vermicompost developed from different sources of organic wastes (feedstock) can be recommended for different crops depending on their nutritional requirements.

**Author Contributions:** Conceptualization, Z.C.; Data curation, W.H. and A.K.; Formal analysis, N.A., N.K.N., A.K. and M.I.K.; Funding acquisition, Z.A., S.B. and M.M.; Investigation, Z.A. and W.H.; Methodology, N.A., N.K.N. and M.I.K.; Resources, M.M.; Supervision, K.B. and Z.C.; Validation, N.A. and N.K.N.; Writing—original draft, S.B.; Writing—review & editing, K.B., Z.C. and M.M.

**Funding:** This work was financially supported by Higher Education Commission of Pakistan (NRPU project # 7527 to Dr. Zubair Aslam and 9017 to Dr. Safdar Bashir). This work was supported by the Internal Grant Agency (IGA) of the Faculty of Economics and Management, Czech University of Life Sciences Prague, grant no. 2019B0011 "Economic analysis of water balance of the current agricultural commodities production mix in the Czech Republic" (Ekonomická analýza vodní bilance stávajícího produkčního mixu zemědělských komodit v ČR) to Dr. Mansoor Maitah.

**Conflicts of Interest**: The authors declare no conflicts of interest.

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
