# Peer review of "Unveiling the Efficiency of Vermicompost Derived from Different Biowastes on Wheat (Triticum aestivum L.) Plant Growth and Soil Health"

_agronomy, doi:10.3390/agronomy9120791_

Round 1

Reviewer 1 Report

This paper compares multiple sources of vermicompost on soil properties and wheat production in an agricultural setting. Properties of the vermicompost sources are monitored throughout the maturation process. Vermicomposts were added to wheat with varying levels of inorganic fertilizer. Effects on soils and wheat were measured, along with aphids. Effects of the different additive treatments are compared for effects and costs versus yield were calculated. This is important research that needs to occur to minimize inorganic fertilizer usage, maintain soil health, and provide beneficial use of organic waste materials.

However, I have some concerns.

First, the English language requires significant editing.

Second, your analyses and associated results and discussion points are flawed. You are presenting results for effects of vermicomposting treatments when the inorganic fertilizer addition was a variable as well. For the comparisons to be valid, you need to separate by inorganic fertilizer level, and then compare the vermicompost treatments within each of those inorganic fertilizer levels. Ideally, treatments without any inorganic fertilizer would be useful, but you can still tease out the different effects of vermicomposts with what you have collected. The data in question are presented in Tables 4 and 5

The abstract presents too much specific information. The main findings should be presented, not every data point that was obtained.

In the Highlights (Lines 51-52), Vermicompost does not need capitalized, and VC should be spelled out.

Throughout the manuscript, you do not need to provide citations and the reference number.

Line 78: What is FYM? This should be spelled out.

Methods Section 2.4 (Page 6): How large were the plots?

Methods Section 2.6 (Page 6): More information is needed here. What were the main effects tested? What variables were analyzed? More information on the analyses were provided in the abstract than are listed here.

Line 178: Be mindful of significant figures. Do not use more than the analytic instrument provides.

Table 3: These fresh materials only had 3% organic matter? I find that difficult to accept.

Page 9: Was any measure of mass loss during the vermicosomposting process collected?

Tables are out of order.

Author Response

The authors are thankful to reviewer for his time and suggestions to improve the manuscript. The responses from the authors are as follows.

This paper compares multiple sources of vermicompost on soil properties and wheat production in an agricultural setting. Properties of the vermicompost sources are monitored throughout the maturation process. Vermicomposts were added to wheat with varying levels of inorganic fertilizer. Effects on soils and wheat were measured, along with aphids. Effects of the different additive treatments are compared for effects and costs versus yield were calculated. This is important research that needs to occur to minimize inorganic fertilizer usage, maintain soil health, and provide beneficial use of organic waste materials.

However, I have some concerns.

First, the English language requires significant editing.

Response: We appreciate reviewer’s comments and thanks for his time in reviewing, Manuscript has been revised and English language has been improved

Second, your analyses and associated results and discussion points are flawed. You are presenting results for effects of vermicomposting treatments when the inorganic fertilizer addition was a variable as well. For the comparisons to be valid, you need to separate by inorganic fertilizer level, and then compare the vermicompost treatments within each of those inorganic fertilizer levels. Ideally, treatments without any inorganic fertilizer would be useful, but you can still tease out the different effects of vermicomposts with what you have collected. The data in question are presented in Tables 4 and 5

Response: We appreciate reviewer’s comments and thanks for his time in reviewing. This was a preliminary study which was conducted to asses the effects of different vermicomposts in combination with inorganic fertilizers and their effects on soil and plant growth. The hypothesis was whether we can reduce the level of recommended NPK using vermicompost in combination and the experiments were conducted to test this hypothesis and different levels of NPK were combined with different vermicompost. In the end we could find the best treatment which performed well in post-harvest analysis of plant and soil as well as based on economic analysis. Field investigations are already in process to solely compare vermicomposts and inorganic fertilizers but this time we focused only on sustainable farming technologies.

The abstract presents too much specific information. The main findings should be presented, not every data point that was obtained.

Response: Abstract is improved

In the Highlights (Lines 51-52), Vermicompost does not need capitalized, and VC should be spelled out.

Response: Highlights were improved and corrected

Throughout the manuscript, you do not need to provide citations and the reference number.

Response: Statements are citied according to needs

Line 78: What is FYM? This should be spelled out.

Response: Corrected

Methods Section 2.4 (Page 6): How large were the plots?

Response: 20 × 19.3 m = 386 m2

Methods Section 2.6 (Page 6): More information is needed here. What were the main effects tested? What variables were analyzed? More information on the analyses were provided in the abstract than are listed here.

Response: The section is revised

Line 178: Be mindful of significant figures. Do not use more than the analytic instrument provides.

Response: The data was rechecked and same figures we observed which are mentioned.

Table 3: These fresh materials only had 3% organic matter? I find that difficult to accept.

Response: We rechecked the data again and found the same %age we can say as all the materials were or organic origin and same %age can be expected.

Page 9: Was any measure of mass loss during the vermicosomposting process collected?

Response: Yes, the loss in weight was observed but it is not mentioned in this paper

Tables are out of order.

Response: Table order has been corrected

Reviewer 2 Report

The authors evaluated the effect of various vermicompost on wheat yield components and aphid control. What comes out from this study is that certain vermicomposts determine economic benefits.

The topic is well on scope with Agronomy. My concerns regard missing information in the Materials and Methods section, the statistical analysis, and the fact that results are not fully discussed. Looking at the objectives and relevance of the study, my recommendation is for a major revision.

Specific comments are listed below.

Highlights:

What is a VC (vermiCompost?)?

Overall, highlights are too long

Abstract:

Too long according to the Journal guidelines. Please, try to shorten it. I found some repetitions.

Also, I do not understand “Minimum aphid population”. It appears out of the blue here.

What do you refer to with “minimum”?

Introduction:

Line 85: Please, add the Latin name of the aphid.

Lines 88-91: I really do not see the link between the emerging problem of aphid and the need for searching effective bio-fertilizers. Please, specify how VC can contribute to affect aphid infestation. This is part of your study hypothesis.

Materials and Methods:

Lines 138-141: Why two ploughings? Were these two performed at different times? Why were sowing density and seed rate so low?

Lines 148-153: At what stages of the VC process did you measure the analytical parameters? See comment related to the Discussions.

Line 153-154: At what phenological stage (or stages) did you record aphid data?

Line 169: Did not you perform the ANOVA test first? The LSD post hoc test is not recommended when you have more than 2 means under comparison because the probability to make the Type I error: it increases dramatically with the increase in the number of the comparisons. With a balanced design, the Ryan Einot Gabriel Welsch – Q or F - test is the most recommended. What was your experimental design scheme?

Discussions:

Lines: 231-240: this part seems to belong to the introduction. Not really a discussion of your results.

I found this part mostly focused on previous studies rather than on your outcome.

Line 259: I did not understand whether you analysed parameters over the vermicompost process or just at the end of the process. To me, it is not evident.

Conclusions:

“and biocontrol agent for aphid attack”: T1 seems to result in similar aphid attack of T2, T5, and T8. Wheat tillers under all the other VC treatments had many more aphids (did you run an ANOVA+post hoc test on these data?).

Author Response

Authors really appreciate reviewers time for their suggestions to improve the manuscript. The comments are addressed and response is given against each comment.

The authors evaluated the effect of various vermicompost on wheat yield components and aphid control. What comes out from this study is that certain vermicomposts determine economic benefits.

The topic is well on scope with Agronomy. My concerns regard missing information in the Materials and Methods section, the statistical analysis, and the fact that results are not fully discussed. Looking at the objectives and relevance of the study, my recommendation is for a major revision.

Specific comments are listed below.

Highlights:

What is a VC (vermiCompost?)?

Response: Yes, it is vermicompost

Overall, highlights are too long

Response: Highlights are revised and shortened

Abstract:

Too long according to the Journal guidelines. Please, try to shorten it. I found some repetitions.

Also, I do not understand “Minimum aphid population”. It appears out of the blue here.

What do you refer to with “minimum”?

Response: Abstract improved and corrected according to suggestions

Introduction:

Line 85: Please, add the Latin name of the aphid.

Response: Added s per suggestions

Lines 88-91: I really do not see the link between the emerging problem of aphid and the need for searching effective bio-fertilizers. Please, specify how VC can contribute to affect aphid infestation. This is part of your study hypothesis.

Response: We hypothesize that the vermicompost will improve microbial community dynamics in soil which will result in synthesis of phytohormones, increase in phosphorus and nitrogen uptake and increase in iron and mineral solubility through chelation growth. Thus, these indirect mechanisms will help to improve plant vigor and develop resistance against aphid infestation

Materials and Methods:

Lines 138-141: Why two ploughings? Were these two performed at different times? Why were sowing density and seed rate so low?

Response: Two cross ploughings were done consecutively to develop normal seedbed. It was normal seed rate 125 kg ha-1 as recommended for normal sowing time in Punjab, Pakistan.

Lines 148-153: At what stages of the VC process did you measure the analytical parameters? See comment related to the Discussions.

Response: Different time intervals were decided to check the changes in VC material during vermicomposting (90 and 180 days)

Line 153-154: At what phenological stage (or stages) did you record aphid data?

Response: The data for aphid population was recorded at milking stage of wheat crop.

Line 169: Did not you perform the ANOVA test first? The LSD post hoc test is not recommended when you have more than 2 means under comparison because the probability to make the Type I error: it increases dramatically with the increase in the number of the comparisons. With a balanced design, the Ryan Einot Gabriel Welsch – Q or F - test is the most recommended. What was your experimental design scheme?

Response: Thanks to reviewer for mentioning this, the data and analysis were rechecked, Dear reviewer we have applied F-test for all the statistical analysis in this paper, and for the treatment means we have LSD test. Those attributes which are significant have lettering, if you say we will provide ANOVA for all attributes

Discussions:

Lines: 231-240: this part seems to belong to the introduction. Not really a discussion of your results.

I found this part mostly focused on previous studies rather than on your outcome.

Response: This section is revised and improved in the revised manuscript

Line 259: I did not understand whether you analysed parameters over the vermicompost process or just at the end of the process. To me, it is not evident.

Response: We showed the changes in physicochemical properties of vermicomposting during different time intervals 90 or 180 days.

Conclusions:

“and biocontrol agent for aphid attack”: T1 seems to result in similar aphid attack of T2, T5, and T8. Wheat tillers under all the other VC treatments had many more aphids (did you run an ANOVA+post hoc test on these data?).

Response: The reviewer is right, and we just observed a secondary benefit of vermicompost to reduce aphid population. Means and standard deviation of the aphid data were calculated, and further studies are in plan to understand real time mechanisms and effects of vermicomposts on insect infestation. The statement is revised and incorporated in the revised manuscript.

Round 2

Reviewer 1 Report

This manuscript is much improved. There are still a few minor changes that would improve the paper.

The mean values for every variable that was measured are still provided in the abstract. The abstract should really focus on the most important findings rather than providing data.

There are a few grammatical errors spread throughout the manuscript but the revision is much improved.

The plot dimensions are provided in the author response but do not appear to be added to the manuscript.

The response regarding separation of inorganic from organic materials is acceptable, but this inability to separate both from the results should be mentioned in the discussion.

Author Response

This manuscript is much improved. There are still a few minor changes that would improve the paper.

The mean values for every variable that was measured are still provided in the abstract. The abstract should really focus on the most important findings rather than providing data.

Authors are thankful to reviewer for their time and valuable comments. The abstract is modified according to the suggestion.

There are a few grammatical errors spread throughout the manuscript, but the revision is much improved.

The manuscript is revised and proofread accordingly.

The plot dimensions are provided in the author response but do not appear to be added to the manuscript.

Plot dimensions are added. Line-151

The response regarding separation of inorganic from organic materials is acceptable, but this inability to separate both from the results should be mentioned in the discussion.

The discussion is modified accordingly line 423-424

Reviewer 2 Report

Authors addressed all the shortcomings I had detected in the first round of revision.

The paper is now ready to be published.

Author Response

Authors are thankful to reviewer for his time and valuable suggestions.